fluid mechanics/mathematical modelling/ mechanical engineering

electrohydrodynamics, reduced-order modelling, data-driven modelling, proper orthogonal decomposition, sparse identification of nonlinear dynamics

**Author for correspondence:**
Yifei Guan
e-mail: yg62@rice.edu

# Sparse nonlinear models of chaotic electroconvection

Yifei Guan[1], Steven L. Brunton[2] and Igor Novosselov[2]

[1]Department of Mechanical Engineering, Rice University, Houston, TX, 77005, USA
[2]Department of Mechanical Engineering, University of Washington, Seattle, WA 98195, USA

YG, 0000-0003-2070-3654; SLB, 0000-0002-6565-5118; IN, 0000-0002-6347-7450

Convection is a fundamental fluid transport phenomenon, where the large-scale motion of a fluid is driven, for example, by a thermal gradient or an electric potential. Modelling convection has given rise to the development of chaos theory and the reduced-order modelling of multiphysics systems; however, these models have been limited to relatively simple thermal convection phenomena. In this work, we develop a reduced-order model for chaotic electroconvection at high electric Rayleigh number. The chaos in this system is related to the standard Lorenz model obtained from Rayleigh–Benard convection, although our system is driven by a more complex three-way coupling between the fluid, the charge density, and the electric field. Coherent structures are extracted from temporally and spatially resolved charge density fields via proper orthogonal decomposition (POD). A nonlinear model is then developed for the chaotic time evolution of these coherent structures using the sparse identification of nonlinear dynamics (SINDy) algorithm, constrained to preserve the symmetries observed in the original system. The resulting model exhibits the dominant chaotic dynamics of the original high-dimensional system, capturing the essential nonlinear interactions with a simple reduced-order model.

## 1. Introduction

In convection, a body force acting on the fluid can lead to the formation of coherent flow structures, with examples including thermal Rayleigh–Bénard convection (RBC) [1–4], surface tension-driven Marangoni effects [5–7], solar magneto-convection [8,9], magnetohydrodynamic convection [10,11], and electroconvection (EC). Quantitative analysis of thermal convection phenomena was first offered by Lord Rayleigh in the early twentieth century [2]. The initially structured convective patterns were subjected to symmetry breaking perturbations and developed into chaotic motion when a critical parameter was

**Table 1.** Abbreviations.

| | |
|---|---|
| $C$ | Charge injection level |
| EC | Electroconvection |
| EHD | Electro-hydrodynamic |
| EK | Electrokinetic |
| EKI | Electro-kinetic instability |
| FBB | Full-way bounce back |
| $Fe$ | Reciprocal of the charge diffusivity coefficient |
| FFT | Fast Fourier transform |
| GPU | Graphics processing unit |
| LBM | Lattice Boltzmann method |
| $M$ | Ratio between hydrodynamic mobility and the ionic mobility |
| NSE | Navier–Stokes equations |
| PC | Principal component |
| POD | Proper orthogonal decomposition |
| $Ra$ | Rayleigh number |
| RBC | Rayleigh-Bénard convection |
| $Re$ | Reynolds number |
| SINDy | Sparse identification of nonlinear dynamics |
| $T$ | Electric Rayleigh number/Taylor number |
| TM | Taylor–Melcher |
| TRT | Two-relaxation time |

reached. Lorenz, Fetter, and Hamilton[1] [12] later discovered a simple reduced-order model for RBC, explaining that the transition to chaotic flow is driven by a thermal gradient, i.e. a two-way coupling between the body force and the fluid motion. This landmark result changed how researchers view the role of numerical simulations and reduced-order modelling in science, and it established the modern field of chaos theory. Similar approaches have been applied to rotating convection or Ferrofluid convection resulting in a three-mode second-order nonlinear model known as the Glukhovsky–Dolzhansky system [13–16]. However, this type of simplified dynamical system model has not yet been demonstrated for convective systems with more complex coupling mechanisms. Electroconvection is a suitable system to extend these analyses, as it involves a nontrivial three-way, multiphysics coupling between the fluid, the electric field, and the charge density. In this work, we demonstrate a modern data-driven approach to identify sparse nonlinear reduced-order models for electroconvection. Our approach, in a way, automates the methods employed by Lorenz, leveraging recent techniques in sparse regression and optimization (see table 1).

The earliest recorded electrohydrodynamic (EHD) experiment dates back to the seventeenth century [17], and the phenomenon of electroconvection in EHD was first reported by G. I. Taylor, while describing the cellular convection in a liquid droplet [18]. Melcher later extended EHD theory to include EC, in what is now known as the Taylor–Melcher (TM) model. Since then, EC has been observed in other systems that exhibit coupling between electrostatic force and fluid motion. The term 'electroconvection' has been used in several different contexts, including electro-kinetic instability (EKI) [19–21] and electro-hydrodynamic (EHD) convection [22–24]. EC patterns have also been observed in liquid crystals when induced by an alternating current [23,25]. In charge-neutral electrokinetic (EK) systems, electroconvection is triggered by the electro-osmotic slip of the electrolyte in the electric double layer at membrane surfaces [19,20,26–36]. In non-equilibrium EHD systems

[1]Margaret Hamilton and Ellen Fetter were graduate students at MIT and assisted Lorenz in programming the LGP-30 computer that was used for simulations. Although Lorenz included them both in the acknowledgements of his papers, by modern standards, Hamilton and Fetter would likely be co-authors on the paper.

**Figure 1.** Summary of nonlinear reduced-order modelling framework for electroconvection. Electroconvection data are taken from the high-fidelity TRT LBM simulation. POD is performed on the charge density field data to extract the dominant spatial coherent structures (modes) and time series (coefficients) for how the mode amplitudes evolve in time. A strong symmetry is observed in the POD coefficient time series. We enforce symmetries in the first three coefficients as constraints on the sparse identification of nonlinear dynamics (SINDy) algorithm. SINDy discovers a sparse nonlinear model to describe the evolution of the mode coefficients, and this model may be used to reconstruct the full charge density field as a linear combination of the corresponding POD modes.

[18,37–44], poorly conductive and leaky dielectric fluid acquires unipolar charge injection at the interface in response to the electric field.

Chaotic flow in EHD convection was first observed and analysed by Atten, Lacroix, and Malraison [38,45]. They characterized two types of behaviours of the power spectra of the intensity fluctuations, identifying two scenarios: dominating viscous force or inertia. They later reported that chaotic charge mixing occurs even at low Reynolds number ($Re$), which in turn can be interpreted as the origin of EC chaos, leading to the onset of hydrodynamic turbulence [46]. The transition to chaos occurs in a variety of convective flow systems at high Rayleigh number ($Ra$), which is the ratio of body forces to viscous forces. Similar to chaotic RBC, chaotic EC is parameterized by a high electric Rayleigh number, also known as the Taylor number ($T$).

The stability of EC systems is now routinely studied with high-fidelity numerical simulations [39,41,42,44,47–52], resulting in large volumes of spatiotemporal data. Despite the high-dimensional nature of these data, many phenomena in electroconvection and other systems may be characterized by the evolution of a few dominant coherent structures [53–55] or *modes*. High-dimensional simulations often obscure this simplicity, making optimization and control tasks prohibitively expensive [56]. Thus, there is a need for accurate and efficient reduced-order models that describe EC phenomena, similar to the Lorenz model for RBC [12]. The Lorenz model was obtained by Galerkin projection of the governing equations onto a dramatically reduced set of three modes, given by the first three Fourier modes proposed by Saltzman [57]. Since Lorenz, this approach has been applied extensively to RBC [58–66]. Instead of extracting these modes manually, modes are now typically extracted automatically, for example via proper orthogonal decomposition (POD) [53,67–70].[2] Galerkin projection of the governing partial differential equation onto POD modes is a more modern approach to obtain a reduced set of ordinary differential equations [53,76–78]. Recently, it was shown that closely related nonlinear reduced-order models could be obtained for fluids purely from measurement data, and without recourse to the governing equations, by applying the sparse identification of nonlinear dynamics (SINDy) algorithm [79–82] to time-series data of POD mode amplitudes. Loiseau *et al.* [80,81] showed that it is also possible to incorporate partially known physics, such as energy conservation, as a constraint in the SINDy regression.

In this work, we demonstrate a data-driven framework to obtain reduced-order models of EC, shown in figure 1. In particular, we examine a non-equilibrium system with unipolar charge injection and an

[2]According to Lumley [69] and Holmes [53], POD was introduced independently at various times by several researchers, including Karhunen [71], Loeve [72], Pugachev [73], Obukhov [74], and Lorenz [75].

external electric field. We first extract dominant coherent structures from DNS of the three-way coupled system, using POD. Strong symmetries are observed in the time evolution of the POD modes, and these symmetries are used as constraints to identify a sparse nonlinear model with SINDy. Surprisingly, we find that the resulting model captures the dominant dynamics using data from the charge density field alone.

# 2. Governing equations, dimensional analysis and simulations

The governing equations for EHD driven flow with unipolar charge injection include the Navier–Stokes equations (NSE) with the electric forcing term $\mathbf{F}_e = -\rho_c \nabla \phi$ in the momentum equation, the charge transport equation and the Poisson equation for electric potential:

$$\nabla \cdot \mathbf{u}^* = 0 \tag{2.1a}$$

$$\rho \frac{D\mathbf{u}^*}{Dt^*} = -\nabla P^* + \mu \nabla^2 \mathbf{u}^* - \rho_c^* \nabla \phi^* \tag{2.1b}$$

$$\frac{\partial \rho_c^*}{\partial t^*} = -\nabla \cdot [(\mathbf{u}^* - \mu_b \nabla \phi^*) \rho_c^* - D_c \nabla \rho_c^*] \tag{2.1c}$$

and

$$\nabla^2 \phi^* = -\frac{\rho_c^*}{\epsilon}, \tag{2.1d}$$

where the asterisk denotes dimensional variables: the two-dimensional velocity field $\mathbf{u}^* = (u_x^*, u_y^*)$, the fluid density $\rho^*$, the static pressure $P^*$, the charge density $\rho_c^*$, and the electric potential $\phi^*$. The parameters are given by the dynamic viscosity $\mu$, the ion mobility $\mu_b$, the ion diffusivity $D_c$, and the electric permittivity $\varepsilon$. The electric force acts as a source term in the momentum equation (2.1b) [83,84]. The system can be non-dimensionalized [50] by introducing four dimensionless parameters [43,44,49]:

$$M = \frac{(\epsilon/\rho)^{1/2}}{\mu_b}, \ \ T = \frac{\epsilon \phi_0}{\mu \mu_b}, \ C = \frac{\rho_0 H^2}{\epsilon \phi_0}, \ \ Fe = \frac{\mu_b \phi_0}{D_c}, \tag{2.2}$$

where $H$ is the distance between the electrodes (two plates infinite in $x$), $\rho_0$ is the injected charge density at the anode, and $\phi_0$ is the voltage difference between the electrodes. In turn, the time $t^*$ can be non-dimensionalized by $H^2/(\mu_b \phi_0)$, $\mathbf{u}^*$—by the ion drift velocity $u_{\text{drift}} = \mu_b \phi_0/H$, $P^*$—by $\rho_0(\mu_b \phi_0)^2/H^2$, $\phi^*$—by $\phi_0$, and the charge density $\rho_c^*$—by $\rho_0$. The physical interpretation of these parameters is as follows: $M$ is the ratio between hydrodynamic mobility and the ionic mobility, $T$ is the ratio between electric force to the viscous force, $C$ is the charge injection level and $Fe$ is the reciprocal of the charge diffusivity coefficient [43,44].

The nondimensionalized form of the governing equations (2.1) is

$$\nabla \cdot \mathbf{u} = 0 \tag{2.3a}$$

$$\frac{D\mathbf{u}}{Dt} = -\nabla P + \frac{M^2}{T} \nabla^2 \mathbf{u} - CM^2 \rho_c \nabla \phi \tag{2.3b}$$

$$\frac{\partial \rho_c}{\partial t} = -\nabla \cdot \left[ (\mathbf{u} - \nabla \phi) \rho_c - \frac{1}{Fe} \nabla \rho_c \right] \tag{2.3c}$$

and

$$\nabla^2 \phi = -C \rho_c \tag{2.3d}$$

where variables without asterisks are dimensionless.

## 2.1. Direct numerical simulations

Several numerical approaches have been developed to study EHD, including finite-difference [39], particle-in-cell [47], and finite-volume methods with the total variation diminishing scheme [41,42]. More recently, the lattice Boltzmann method (LBM) was used to predict the linear and finite-amplitude stability criteria of the subcritical bifurcation in the EC flow [44,48]. A segregated solver was proposed that combines a two-relaxation time (TRT) LBM modelling of the fluid and charge transport and a Fast Fourier Transform (FFT) Poisson solver for the electrical field [49,50].

Due to its computational efficiency, direct numerical simulations in this study are performed using the TRT LBM approach to solve the transport equations for fluid flow and charge density, coupled to a fast Poisson solver for the electric potential [49,50,85]. The numerical method is implemented in C++ using CUDA GPU computing. A spatial resolution of $122 \times 100$ is used, balancing accuracy with efficiency; the number of threads in the $x$-direction in each GPU block is equal to the grid resolution

**Table 2.** Boundary conditions for the numerical simulations.

| boundary | macro-variables conditions | meso-variables conditions |
|---|---|---|
| x-direction | periodic | periodic |
| Upper plate | $\mathbf{u} = 0$, $\phi = 0$, $\partial \rho_c / \partial y = 0$ | LBM FBB scheme for $f_i$; $\partial g_i / \partial y = 0$ |
| Lower plate | $\mathbf{u} = 0$, $\phi = \phi_0$, $\rho_c = \rho_0$ | LBM FBB for both $f_i$ and $g_i$ |

in $x$, and the number of GPU blocks in the $y$-direction is equal to the grid resolution in $y$. FFT and IFFT operations are performed using the cuFFT library [86]. All variables are computed with double precision to reduce the truncation error. The numerical method was shown to be second-order accurate in both time and space.

The non-dimensional parameters used in this study are $C = 10$, $M = 10$, $T = 312.5$, and $Fe = 4000$. $C = 10$ corresponds to a strong charge injection; $M = 10$ and $Fe = 4000$ correspond to the typical values of mobility and charge diffusivity of a dielectric liquid [43,87]. $T$ governs the viscosity of the flow, where $T = 312.5$ is near the onset of chaos. The macroscopic and mesoscopic boundary conditions are specified in table 2. The no-slip boundary conditions are applied at both electrodes for fluid flow. A constant charge density at the anode (lower plate) represents a unipolar injection; a zero-diffusive flux condition $\nabla \rho_c = 0$ at the cathode (upper plate) represents an outflowing current. A constant electric potential is applied at the anode; the cathode is grounded ($\phi = 0$). At mesoscale (LBM scale), the discrete distribution function of velocity $f_i(\mathbf{x}, t)$ and charge density $g_i(\mathbf{x}, t)$ are used. The details on the transformations between macro-variables ($\mathbf{u}$, $\rho_c$) and meso-variables ($f_i$, $g_i$) can be found in recent publications [49,50]. The LBM full-way bounce-back (FBB) scheme is used for the Dirichlet (no-slip) boundary conditions for the fluid flow and charge density at the lower plate [44,50]. The $g_i$ Neumann boundary condition is set as a current outlet boundary condition for charge density transport [44,49]. The abbreviations are listed in table 1.

Figure 2a shows an illustration of the charge density from the electroconvection simulation, and figure 2b shows the unsteady nature of the ionic convection leading to electric field variation and consequently the unsteady flow patterns. This is apparent by examining the momentum equation, equation (2.3b), as the variance in the source term drives the instability in the flow if the viscosity term cannot dampen the flow fluctuations. For the chosen parameters $C = 10$, $M = 10$, $T = 312.5$, and $Fe = 4000$, the flow is driven by a strong charge injection with a high electric force to viscous force ratio ($T$). The irregular profiles of charge density represent states with an imbalance of electric force and viscous force.

# 3. Coherent structures and symmetries

We now extract dominant coherent structures and symmetries from the time-resolved DNS data from above. In particular, we use POD to extract an orthogonal set of spatial modes from the charge density field, along with time series for how these mode amplitudes evolve in time; these modes and amplitudes are shown in figure 2c,d. A spatially and temporally resolved data set is used with a time step of $\Delta t = 0.005$ from $t = 0$ to $t = 1000$. The zeroth POD mode (PC0) represents the charge density mean-field and does not vary with time. The first POD mode (PC1) is strongly periodic in both space and time, with periodic structures corresponding to the two up-drifting ion channels between charge void regions, as shown in figure 2a. During the simulation, the ion channels oscillate and can merge and separate, as shown in figure 2b. The second and third POD modes (PC2,3) are a phase-shifted mode pair that contribute to the oscillation of the ion channels. Higher modes are not obviously paired, although they do correspond to spatial and temporal harmonics. POD was also computed on the other spatial fields, and structures were qualitatively similar.

Figure 3 shows the trajectories of the first three POD coefficients ($a_1$, $a_2$ and $a_3$) and their projections onto two-dimensional planes. These trajectories exhibit several strong symmetries, so that the system may be viewed as invariant with respect to the following transformations:

$$[a_1, a_2, a_3] \leftrightarrow [a_1, -a_2, a_3] \leftrightarrow [a_1, a_2, -a_3] \leftrightarrow [a_1, -a_2, -a_3] \leftrightarrow [-a_1, a_3, a_2] \leftrightarrow [-a_1, -a_3, -a_2]. \quad (3.1)$$

These symmetries in the original system will be enforced in our sparse nonlinear models.

Note that POD and Galerkin projection have been widely applied to RBC, although analyses are limited for EC. Sirovich *et al.* applied POD to RBC to investigate the scaling of POD modes to

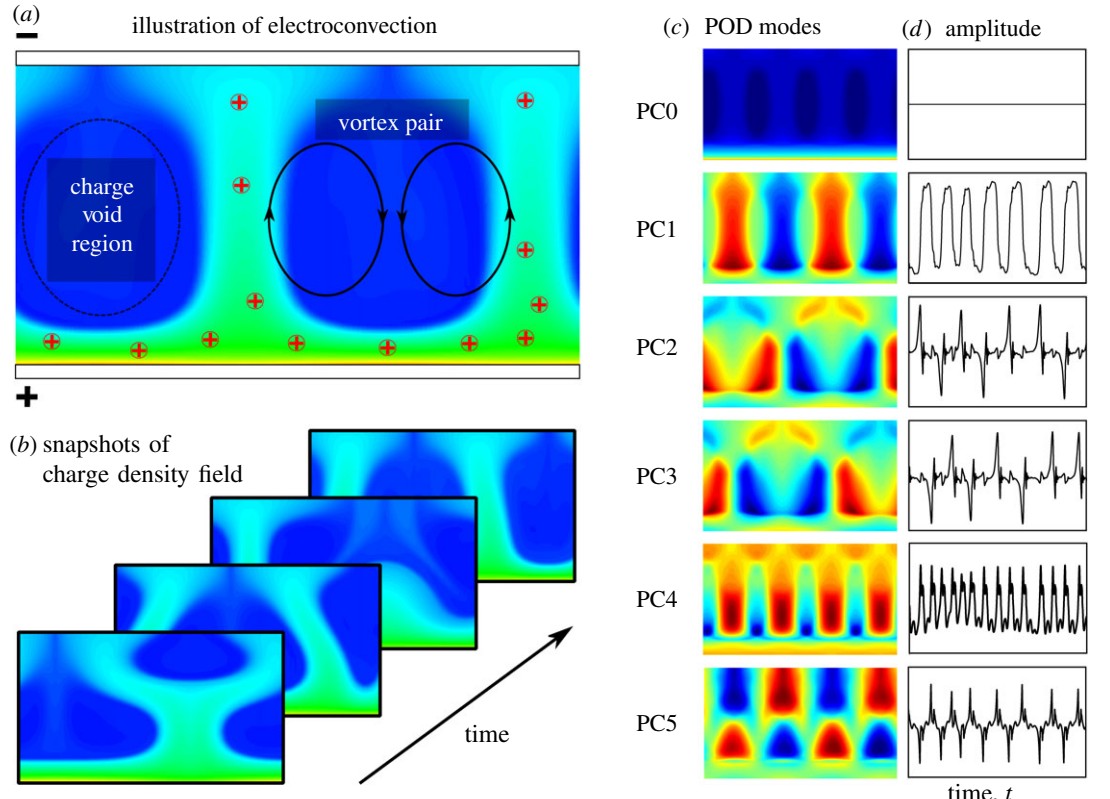

**Figure 2.** Electroconvection simulation data and proper orthogonal decmoposition. (*a*) Illustration of electroconvection of a dielectric fluid between parallel electrodes under strong unipolar injection; (*b*) snapshots of charge density field used for POD analysis; (*c,d*) mean charge density field and first five POD modes and amplitudes in time. (PC, principal component; PC0 is the mean field)

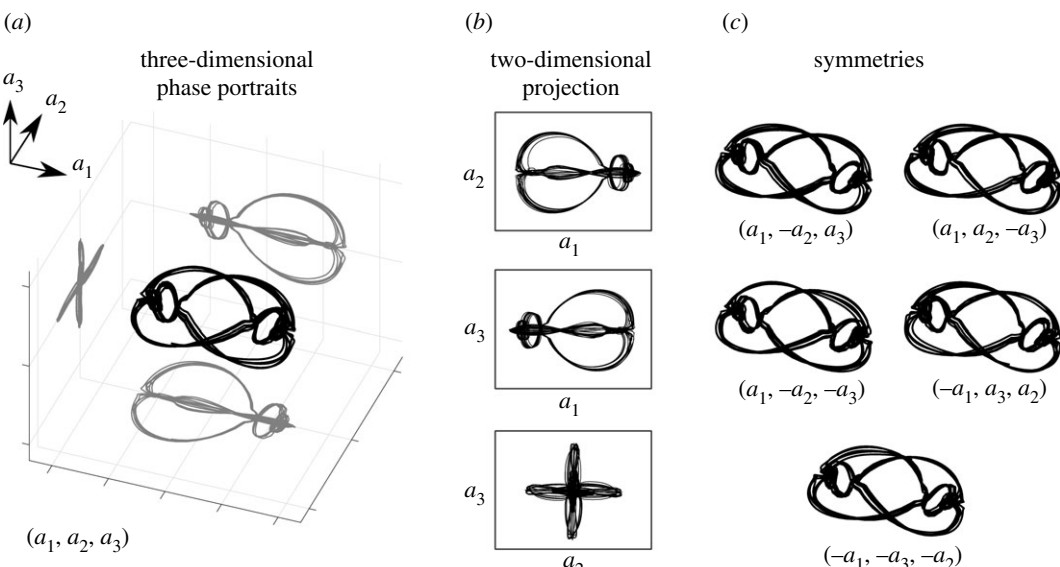

**Figure 3.** Symmetries in POD mode coefficients. (*a*) three-dimensional phase portraits of the first three POD modes; (*b*) two-dimensional projection of the three-dimensional phase portraits onto two-dimensional planes; (*c*) symmetry transformations of the three-dimensional phase portraits, which imply symmetry in the underlying dynamical system.

Rayleigh number [58] and to specify the chaotic RBC with Liapunov and Karhunen–Loeve dimensions [59–61]. Low-dimensional dynamics and data compression were also considered [62]. To further investigate the low-dimensional model of RBC, Bailon-Cuda *et al.* performed Galerkin projection onto POD modes from DNS of turbulent RBC and found that a few hundred POD modes are required to

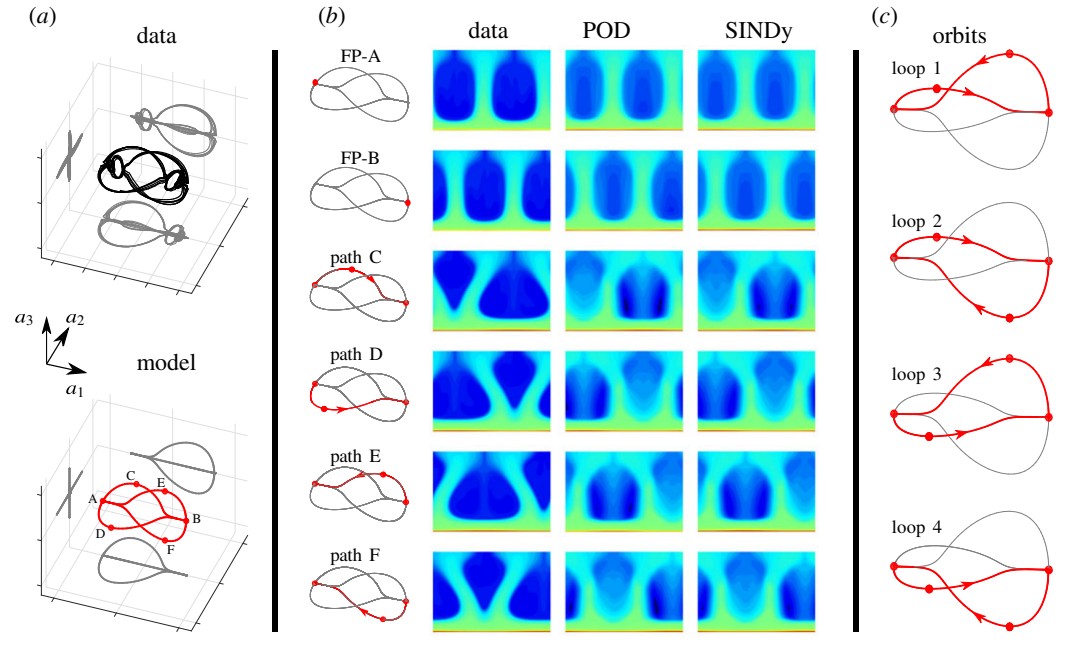

**Figure 4.** Comparison of full simulation and SINDy model for electroconvection. (*a*) Three-dimensional phase portraits of data from numerical simulation and from SINDy model; (*b*) charge density fields and their reconstructions by POD modes and SINDy model. The flow is characterized by two fixed points, FP-A and FP-B, and there are four paths that are taken between these fixed points, labelled C-F. (*c*) Four different loops are formed by the four paths (C-F).

qualitatively reproduce the large-scale turbulent convection [63–65]. Recently, Cai *et al.* developed a closure scheme for two-dimensional RBC at high *Ra* number with POD-Galerkin models [66].

# 4. Sparse nonlinear reduced-order models

We now demonstrate the identification of a parsimonious nonlinear model for EC using the sparse identification of nonlinear dynamics (SINDy) [79] approach; results are summarized in figure 4. SINDy has been widely applied for model identification in a variety of applications, including chemical reaction dynamics [88], nonlinear optics [89], fluid dynamics [80–82,90,91] and turbulence modelling [92,93], plasma convection [94], numerical algorithms [95], and structural modelling [96], among others [97–99]. Of particular note are its uses in identifying Lorenz-like dynamics from a thermosyphon simulation by Loiseau [82] and to identify a model for a nonlinear magnetohydrodynamic plasma system by Kaptanoglu *et al.* [100]. It has also been extended to handle more complex modelling scenarios such as partial differential equations [101,102], systems with inputs or control [103], to enforce physical constraints [80], to identify models from corrupt or limited data [104,105] and ensembles of initial conditions [106], and extending the formulation to include integral terms [107,108], tensor representations [109,110], deep autoencoders [111], and stochastic forcing [112,113]; an open-source software package, PySINDy, has been developed to integrate a number of these innovations [114]. We will enforce the symmetries observed above as constraints, as in Loiseau and Brunton [80].

We apply SINDy to develop a sparse reduced-order model of electroconvection, in particular the evolution of the POD coefficients $a_1(t)$, $a_2(t)$ and $a_3(t)$. First, we construct a data matrix $\mathbf{X}$ whose columns are time-series of $a_1$, $a_2$, and $a_3$, along with the corresponding matrix of time derivative, $\dot{\mathbf{X}}$:

$$\mathbf{X} = \begin{bmatrix} a_1(t_1) & a_2(t_1) & a_3(t_1) \\ a_1(t_2) & a_2(t_2) & a_3(t_2) \\ \vdots & \vdots & \vdots \\ a_1(t_m) & a_2(t_m) & a_3(t_m) \end{bmatrix} \quad \text{and} \quad \dot{\mathbf{X}} = \begin{bmatrix} \dot{a}_1(t_1) & \dot{a}_2(t_1) & \dot{a}_3(t_1) \\ \dot{a}_1(t_2) & \dot{a}_2(t_2) & \dot{a}_3(t_2) \\ \vdots & \vdots & \vdots \\ \dot{a}_1(t_m) & \dot{a}_2(t_m) & \dot{a}_3(t_m) \end{bmatrix}. \qquad (4.1)$$

Based on the data in $\mathbf{X}$, a library of candidate nonlinear functions $\Theta(\mathbf{X})$ is constructed, containing functions that may describe the observed dynamics. This library is only limited by the user's

imagination and may be guided by terms present in the overarching partial differential equation. For fluid systems, polynomials have been quite effective:

$$\Theta(\mathbf{X}) = [\mathbf{1} \ \mathbf{X} \ \mathbf{X}^2 \cdots \mathbf{X}^d], \tag{4.2}$$

where the matrix $\mathbf{X}^d$ denotes a matrix with column vectors given by all possible time series of $d$th degree polynomials in the state $\mathbf{x} = [a_1 \ a_2 \ a_3]$.

SINDy develops reduced-order models by selecting the fewest columns of $\Theta(\mathbf{X})$ that add up in a linear combination to describe the derivatives in $\dot{\mathbf{X}}$:

$$\dot{\mathbf{X}} = \Theta(\mathbf{X})\Xi, \tag{4.3}$$

where $\Xi$ is the sparse coefficient matrix that denotes which columns of $\Theta$ are active, and hence, which terms are active in the dynamics. A parsimonious model will provide an accurate fit with as few terms as possible in $\Xi$. Such a model can be developed using a convex regression:

$$\Xi_k = \mathrm{argmin}_{\Xi'_k} \parallel \dot{\mathbf{X}}_k - \Theta(\mathbf{X})\Xi'_k \parallel_2 + \lambda \parallel \Xi'_k \parallel_1 , \tag{4.4}$$

where $\Xi_k$ denotes the $k$th column of $\Xi$. There are several algorithms to obtain a sparse model $\Xi$, and we use the sequential-thresholded least squares (STLS) algorithm [79]. Loiseau and Brunton [80] showed that it is also possible to incorporate known linear constraint equations in this regression:

$$\min_{\Xi} \parallel \dot{\mathbf{X}} - \Theta(\mathbf{X})\Xi \parallel_2^2 + \lambda \parallel \Xi \parallel_1 \tag{4.5a}$$

and

$$\text{subject to } \mathbf{C}\boldsymbol{\xi} = \mathbf{d} \tag{4.5b}$$

where $\boldsymbol{\xi} = \Xi(:)$ is the vectorized form of the sparse matrix of coefficients, and $\mathbf{C}\boldsymbol{\xi} = \mathbf{d}$ are linear equality constraints, which can be used to enforce symmetries and other known constraints, such as energy conservation. We perform SINDy on the POD coefficients $a_i$ for $i = 1, 2, 3$ with the symmetries observed from the data shown in figure 3.

## 4.1. Enforcing symmetries in SINDy

As shown in figure 3, the trajectories remain nearly unchanged under the symmetry transformations in equation (3.1). These symmetries imply various constraints on the coefficients in $\Xi$, resulting in several terms that are zero and several other terms that must be related across the dynamics; these constraints are summarized in table 3.

The symmetry constraints are determined as follows. First, we rewrite the unknown system as follows

$$\dot{a}_1 = \Theta(a_1, a_2, a_3)\Xi_1, \tag{4.6a}$$
$$\dot{a}_2 = \Theta(a_1, a_2, a_3)\Xi_2, \tag{4.6b}$$
and
$$\dot{a}_3 = \Theta(a_1, a_2, a_3)\Xi_3. \tag{4.6c}$$

For the first row, equation (4.6a), the equation is invariant to switching the sign of $a_2$ and/or $a_3$. Thus, the coefficients of the terms with odd powers of either $a_2$ or $a_3$ must vanish ($\xi_{ij}$ corresponding to $a_1^l a_2^m a_3^n$ where at least one of $m$ and $n$ is odd). From the observed symmetry $[a_1, a_2, a_3] \leftrightarrow [-a_1, a_3, a_2]$, the coefficients $\xi_{ij}$ corresponding to $a_1^l a_2^m a_3^n$ should be set to zero where $l$ is even and non-constrained where $l$ is odd. The rest of the $\xi_{ij}$ correspond to $a_1^l a_2^m a_3^n$ where both $m$ and $n$ are even numbers, namely $a_2 a_2$, $a_3 a_3$, $a_1 a_2 a_2$, $a_1 a_3 a_3$. When $a_1$ is replaced by $-a_1$, $a_2$ and $a_3$ must be swapped for the equation to remain unchanged, so the coefficients $\xi_{ij}$ for $a_2 a_2$ and $a_3 a_3$ should be equal and opposite and those for $a_1 a_2 a_2$ and $a_1 a_3 a_3$ should be identical.

The coefficients in the second and third rows, equations (4.6b) and (4.6c), should be considered together. From the symmetry transformations $a_2 \leftrightarrow -a_2$ and $a_3 \leftrightarrow -a_3$, the coefficients $\xi_{ij}$ corresponding to $a_1^l a_2^m a_3^n$ in the equation of $\dot{a}_2$ and those corresponding to $a_1^l a_2^n a_3^m$ in the equation of $\dot{a}_3$ vanish when $m$ is even or $n$ is odd. The rest of the coefficients $\xi_{ij}$ in equation (4.6b) correspond to $a_1^l a_2^m a_3^n$ where $m$ is odd and $n$ is even, namely $a_2$, $a_1 a_2$, $a_1 a_1 a_2$, $a_2 a_2 a_2$, $a_2 a_3 a_3$. Similarly, the rest of the coefficients $\xi_{ij}$ in equation (4.6c) correspond to $a_1^l a_2^n a_3^m$ where $n$ is even and $m$ is odd, namely $a_3$, $a_1 a_3$, $a_1 a_1 a_3$, $a_2 a_2 a_3$, $a_3 a_3 a_3$. From the observed symmetry $[a_1, a_2, a_3] \leftrightarrow [-a_1, a_3, a_2]$, the coefficients $\xi ij$ corresponding to $a_1^l a_2^m a_3^n$ in equation (4.6b) and those corresponding to $a_1^l a_2^n a_3^m$ in equation (4.6c) should be identical where $l$ is even, and equal and opposite where $l$ is odd.

**Table 3.** Structure of the $\Xi$ matrix.

| | $a_1$ | $a_2$ | $a_3$ | $a_1^2$ | $a_1a_2$ | $a_1a_3$ | $a_2^2$ | $a_2a_3$ | $a_3^2$ | $a_1^3$ | $a_1^2a_2$ | $a_1^2a_3$ | $a_1a_2^2$ | $a_1a_2a_3$ | $a_1a_3^2$ | $a_2^3$ | $a_2^2a_3$ | $a_2a_3^2$ | $a_3^3$ |
|---|---|---|---|---|---|---|---|---|---|---|---|---|---|---|---|---|---|---|---|
| $\dot{a}_1$ | $\xi_1$ | | | | | | $\xi_2^*$ | | $-\xi_2$ | $\xi_3$ | | | $\xi_4$ | | $\xi_4$ | | | | |
| $\dot{a}_2$ | | $\xi_5$ | | | $-\xi_2^*$ | | | | | | $\xi_6$ | | | | | $\xi_7$ | | $\xi_8$ | |
| $\dot{a}_3$ | | | $\xi_5$ | | | $\xi_2$ | | | | | | $\xi_6$ | | | | | $\xi_8$ | | $\xi_7$ |

We explicitly enforce these constraints in the SINDy regression. Each row of table 3 represents each row of the equation (4.3). The blank entries represent zeros, and each $\xi_i$ represents one value, i.e. the entries with the same $\xi_i$ should have the same values. To enforce the energy constraints as described by Loiseau [80], we set the coefficients with asterisk equal and opposite in table 3.

The system is therefore

$$\dot{a}_1 = \xi_1 a_1 + \xi_2 a_2^2 - \xi_2 a_3^2 + \xi_3 a_1^3 + \xi_4 a_1 a_2^2 + \xi_4 a_1 a_3^2 \tag{4.7a}$$

$$\dot{a}_2 = \xi_5 a_2 - \xi_2 a_1 a_2 + \xi_6 a_1^2 a_2 + \xi_7 a_2^3 + \xi_8 a_2 a_3^2 \tag{4.7b}$$

and

$$\dot{a}_3 = \xi_5 a_3 + \xi_2 a_1 a_3 + \xi_6 a_1^2 a_3 + \xi_8 a_2^2 a_3 + \xi_7 a_3^3. \tag{4.7c}$$

To further take advantage of the symmetry, we augment the data with all symmetry-transformed copies of the data:

$$\mathbf{X} = [a_1, a_1, a_1, a_1, -a_1, -a_1, a_2, -a_2, a_2, -a_2, a_3, -a_3, a_3, a_3, -a_3, -a_3, a_2, -a_2]. \tag{4.8}$$

For the mildly chaotic EC system in a dielectric liquid driven by a strong charge injection, specified by $C = 10$, $M = 10$, $T = 312.5$, and $Fe = 4000$, with the fractal dimension approximately equal to 2.5 and Lyapunov exponent approximately equal to 0.04 (see ref. [115] for more details on the complete route to the chaos of two-dimensional EC flow), the resulting sparse nonlinear model is

$$\dot{a}_1 = 0.77 a_1 + 0.57 a_2^2 - 0.57 a_3^2 - 0.017 a_1^3 + 0.043 a_1 a_2^2 + 0.043 a_1 a_3^2 \tag{4.9a}$$

$$\dot{a}_2 = -1.54 a_2 - 0.57 a_1 a_2 + 0.0059 a_1^2 a_2 - 0.014 a_2^3 + 0.18 a_2 a_3^2 \tag{4.9b}$$

and

$$\dot{a}_3 = -1.54 a_3 + 0.57 a_1 a_3 + 0.0059 a_1^2 a_3 + 0.18 a_2^2 a_3 - 0.014 a_3^3. \tag{4.9c}$$

## 4.2. Reconstruction of the charge density field

The results from the SINDy model are depicted in figure 4. The reduced-order model accurately captures the qualitative behaviour of the system, characterized by four paths connecting two fixed points, denoted A and B in the figure. The system pseudo-randomly chooses each of the two paths starting at a given fixed point, resulting in the four orbits shown in figure 4c; this behaviour closely matches the behaviour of the full high-dimensional system. Further, the SINDy model may be used to recombine POD modes to approximate the high-dimensional dynamics around these fixed points and connecting paths. The SINDy reconstruction faithfully captures the truncated POD approximation of the true system; thus, a better model cannot be obtained without including more POD modes in the expansion.

# 5. Conclusion and discussion

In this work, we developed a data-driven reduced-order model for chaotic electroconvection. Our approach is reminiscent of that employed by Lorenz to model Rayleigh–Bénard convection, although dimensionality reduction and nonlinear model identification procedures are automated and do not require recourse to the governing equations. In particular, we investigate spatiotemporal data from direct numerical simulations of the three-way coupling between the fluid, charge density and electric field that give rise to chaotic electroconvection. We extract coherent structures from the charge density field using POD, and we develop a nonlinear model for how the POD mode amplitudes evolve in time using the SINDy modelling framework. Critically, we identify several symmetries in the time-series data, which we later enforce as constraints in the SINDy algorithm, dramatically reducing the search space of candidate models. The resulting model faithfully captures the dominant behaviour of the leading POD modes, including chaotic switching in the time series that corresponds to chaotic convection in the full field. Because the nonlinear model is sparse and is formulated in terms of a few key spatial modes, it is interpretable by construction and may be less prone to overfitting compared to deep learning. Thus, we demonstrate the ability of automated model discovery techniques to uncover interpretable dynamical systems models for convective phenomena with more complex three-way coupling than the original Lorenz model of RBC.

There are a number of natural extensions and future directions suggested by this work. Similar to the Hopf–Rayleigh number of the Lorenz system [116], the electric Hopf–Rayleigh number determining the onset of electroconvection chaos can be identified by a linear stability analysis of the nonlinear system investigated here. Additionally, three-dimensional flows and experimental measurements would

provide challenging tests to the methodology and may point out limitations that require further development. Although the present work only considered data from the charge density, incorporating information from the three fluid, electric, and charge density fields is the subject of ongoing research. This will require developing a dimensionally consistent POD, as has been done for compressible flows [117] and magnetohydrodynamics [100]. This modelling framework is also amenable to including the effect of varying parameters, which could result in a parameterized nonlinear model that would enable more sophisticated design optimization and the prediction of bifurcation events; however, incorporating a parameter into the model increases the size of the library of candidate functions and requires more training data. Increasing the number of POD modes in the model will also likely increase the fidelity of the prediction, for example enabling the modelling of subtle secondary effects in the data. However, when additional POD modes are added, it will be necessary to automatically extract symmetries and determine what constraints these symmetries impose on the candidate library. This challenge fits into the broader literature of symmetry reduction [118]. Finally, using these models for design, optimization, and control will be a true test of the model.

Competing interests. We declare we have no competing interests.
Data accessibility. This paper contains no experimental data. Data and relevant code for this research work are stored in GitHub: https://github.com/gyf135/EHD-convection-2D and have been archived within the Zenodo repository: https://doi.org/10.5281/zenodo.5083306.
Authors' contributions. YG, SLB and IN conceived of the work. YG performed the research. SLB provided guidance on modelling. All authors analysed the results and drafted the manuscript.
Competing interests. We declare we have no competing interests.
Funding. SLB acknowledges funding from the Army Research Office (ARO W911NF-19-1-0045). The authors would also like to acknowledge fruitful discussions with Jared Callaham, Alan Kaptanoglu, Nathan Kutz and Jean-Christophe Loiseau. This work was partially supported by the National Institutes of Health (grant nos. NIBIB U01 EB021923, NIBIB R42ES026532 subcontract to UW).

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
