## [Peer Review File · Royal Society Open Science]

Review History

RSOS-202367.R0 (Original submission)

Review form: Reviewer 1

Is the manuscript scientifically sound in its present form?

Yes

Are the interpretations and conclusions justified by the results?

Yes

Is the language acceptable?

Yes

Do you have any ethical concerns with this paper?

No

Have you any concerns about statistical analyses in this paper?

No

Recommendation?

Accept as is

Comments to the Author(s)

The authors propose a pioneering POD model of chaotic electroconvection from a high-fidelity simulation.

The simplicity of the model is achieved by a masterfully performed sophisticated reduced-order modeling (ROM) approach starting with the evolution equations, choosing the right snapshot data, performing POD, enforcing all symmetries and performing sparse regression for human-interpretability.

The work can be expected to inspire the quest for many similar future ROM of multi-physics phenomena and is certainly certainly appropriate for the Royal Society Open Science audience. Moreover, the methods and results are excellently written and illustrated making the manuscript a pleasure to read.

My small list of suggestions is a testimony to this statement.

Summarizing, I enthusiastically recommend publication already in its current form.

As minor suggestion, I propose that the authors clarify the meaning of << chaotic behaviour >>. Does this imply a strange attractor with fractal dimension and positive Lyapunov exponent like for the mentioned Lorenz model or do the authors mean << complex behaviour >>, e.g. with several frequencies, mode switching etc.

A second suggestion is a table of abbreviations.

This is left to the authors discretion.

Review form: Reviewer 2 (Pradeep Siddheshwar)

Is the manuscript scientifically sound in its present form?

Yes

Are the interpretations and conclusions justified by the results?

Yes

Is the language acceptable?

Yes

Do you have any ethical concerns with this paper?

No

Have you any concerns about statistical analyses in this paper?

No

Recommendation?

Accept with minor revision (please list in comments)

Comments to the Author(s)

1. Chaotic electroconvection is dealt with in the paper by considering a reduced-order model that involves three-way coupling. Such situations are seen not only in electroconvection but also in ferroconvection. There are innumerable works that have considered such couplings and reduced-order models. The models go by the name of Glukhovskiy–Dolzhanov system and differ from the classical Lorenz model in being nonlinear in all the three equations and having a three-way coupling in the amplitudes. Works have also appeared wherein this third-order system has been

reduced further to a one-dimensional system using the idea of a center manifold. The authors may contextualize their work by mentioning about such works.

2. Onset of chaotic electroconvection may be discussed by considering the Hopf-Rayleigh number, r_H . A third-order reduced system should not be difficult to handle in order to obtain r_H .

Decision letter (RSOS-202367.R0)

Dear Dr Guan

On behalf of the Editors, we are pleased to inform you that your Manuscript RSOS-202367 "Sparse nonlinear models of chaotic electroconvection" has been accepted for publication in Royal Society Open Science subject to minor revision in accordance with the referees' reports. Please find the referees' comments along with any feedback from the Editors below my signature.

Please submit your revised manuscript and required files (see below) no later than 7 days from today's (ie 01-Jul-2021) date. Note: the ScholarOne system will 'lock' if submission of the revision is attempted 7 or more days after the deadline. If you do not think you will be able to meet this deadline please contact the editorial office immediately.

on behalf of Miles Padgett (Subject Editor)
openscience@royalsociety.org

Reviewer comments to Author:

Reviewer: 1

Comments to the Author(s)

The authors propose a pioneering POD model of chaotic electroconvection from a high-fidelity simulation.

The simplicity of the model is achieved by a masterfully performed sophisticated reduced-order modeling (ROM) approach starting with the evolution equations, choosing the right snapshot data, performing POD, enforcing all symmetries and performing sparse regression for human-interpretability.

The work can be expected to inspire the quest for many similar future ROM of multi-physics phenomena and is certainly certainly appropriate for the Royal Society Open Science audience.

Moreover, the methods and results are excellently written and illustrated making the manuscript a pleasure to read.

My small list of suggestions is a testimony to this statement.

Summarizing, I enthusiastically recommend publication already in its current form.

As minor suggestion, I propose that the authors clarify the meaning of << chaotic behaviour >>. Does this imply a strange attractor with fractal dimension and positive Lyapunov exponent like for the mentioned Lorenz model or do the authors mean << complex behaviour >>, e.g. with several frequencies, mode switching etc.

A second suggestion is a table of abbreviations.

This is left to the authors discretion.

Reviewer: 2

Comments to the Author(s)

1. Chaotic electroconvection is dealt with in the paper by considering a reduced-order model that involves three-way coupling. Such situations are seen not only in electroconvection but also in ferroconvection. There are innumerable works that have considered such couplings and reduced-order models. The models go by the name of Glukhovsky-Dolzhansky system and differ from the classical Lorenz model in being nonlinear in all the three equations and having a three-way coupling in the amplitudes. Works have also appeared wherein this third-order system has been reduced further to a one-dimensional system using the idea of a center manifold. The authors may contextualize their work by mentioning about such works.

2. Onset of chaotic electroconvection may be discussed by considering the Hopf-Rayleigh number, r_H . A third-order reduced system should not be difficult to handle in order to obtain r_H .

===PREPARING YOUR MANUSCRIPT===

Please ensure that you include an acknowledgements' section before your reference list/bibliography. This should acknowledge anyone who assisted with your work, but does not

qualify as an author per the guidelines at <https://royalsociety.org/journals/ethics-policies/openness/>.

===PREPARING YOUR REVISION IN SCHOLARONE===

- Ensure that your data access statement meets the requirements at <https://royalsociety.org/journals/authors/author-guidelines/#data>. You should ensure that you cite the dataset in your reference list. If you have deposited data etc in the Dryad repository, please only include the 'For publication' link at this stage. You should remove the 'For review' link.
- If you are requesting an article processing charge waiver, you must select the relevant waiver option (if requesting a discretionary waiver, the form should have been uploaded at Step 3 'File upload' above).
- If you have uploaded ESM files, please ensure you follow the guidance at <https://royalsociety.org/journals/authors/author-guidelines/#supplementary-material> to include a suitable title and informative caption. An example of appropriate titling and captioning may be found at https://figshare.com/articles/Table_S2_from_Is_there_a_trade-off_between_peak_performance_and_performance_breadth_across_temperatures_for_aerobic_sc_ope_in_teleost_fishes_/3843624.

Author's Response to Decision Letter for (RSOS-202367.R0)

See Appendices A & B.

Decision letter (RSOS-202367.R1)

Dear Dr Guan,

I am pleased to inform you that your manuscript entitled "Sparse nonlinear models of chaotic electroconvection" is now accepted for publication in Royal Society Open Science.

on behalf of Prof Miles Padgett (Subject Editor)
openscience@royalsociety.org

Appendix A

RESPONSE TO REVIEW COMMENTS

Manuscript: Sparse nonlinear models of chaotic electroconvection

Referee 1:

The authors propose a pioneering POD model of chaotic electroconvection from a high-fidelity simulation. The simplicity of the model is achieved by a masterfully performed sophisticated reduced-order modeling (ROM) approach starting with the evolution equations, choosing the right snapshot data, performing POD, enforcing all symmetries and performing sparse regression for human-interpretability. The work can be expected to inspire the quest for many similar future ROM of multi-physics phenomena and is certainly appropriate for the Royal Society Open Science audience. Moreover, the methods and results are excellently written and illustrated making the manuscript a pleasure to read. My small list of suggestions is a testimony to this statement. Summarizing, I enthusiastically recommend publication already in its current form.

As minor suggestion, I propose that the authors clarify the meaning of “chaotic behaviour”. Does this imply a strange attractor with fractal dimension and positive Lyapunov exponent like for the mentioned Lorenz model or do the authors mean “complex behaviour”, e.g., with several frequencies, mode switching etc.

A second suggestion is a table of abbreviations. This is left to the authors discretion.

Response:

The authors are very grateful for the referee’s very positive assessment of our work, and we also appreciate the helpful suggestions. The following is the response to the suggestions.

1. The “chaotic behaviour” means a strange attractor with fractal dimension approximately equal to 2.5 and Lyapunov exponent approximately equal to 0.04. The following statement is added to the manuscript to further supplement the details and clarify the terms:

Section 4(a): ...with the fractal dimension approximately equal to 2.5 and Lyapunov exponent approximately equal to 0.04 (see Ref. (115) for more details on the complete route to the chaos of 2D EC flow).

2. An Abbreviations table is included as per the reviewer’s suggestion.

Reference (numeric same as the main text):

[115] Wang Q, Guan Y, Huang J, Wu J. 2021 Chaotic electro-convection flow states of a dielectric liquid between two parallel electrodes. *European Journal of Mechanics- B/Fluids*.

Appendix B

RESPONSE TO REVIEW COMMENTS

Manuscript: Sparse nonlinear models of chaotic electroconvection

Referee 2:

1. Chaotic electroconvection is dealt with in the paper by considering a reduced-order model that involves three-way coupling. Such situations are seen not only in electroconvection but also in ferroconvection. There are innumerable works that have considered such couplings and reduced-order models. The models go by the name of Glukhovsky–Dolzhansky system and differ from the classical Lorenz model in being nonlinear in all the three equations and having a three-way coupling in the amplitudes. Works have also appeared wherein this third-order system has been reduced further to a one-dimensional system using the idea of a center manifold. The authors may contextualize their work by mentioning about such works.

2. Onset of chaotic electroconvection may be discussed by considering the Hopf-Rayleigh number, r_H . A third-order reduced system should not be difficult to handle in order to obtain r_H .

Response:

The authors appreciate the suggestions by the reviewer. The following is the response to the suggestions.

1. The following statement is added to the manuscript to contextualize the work further:

Introduction: Similar approaches have been applied to rotating convection or Ferrofluid convection resulting in a three-mode second-order nonlinear model known as the Glukhovsky–Dolzhansky system [13; 14; 15; 16].

2. In the direct numerical simulation, the electric Hopf-Rayleigh number is determined as $T=310$ at the strong charge injection condition as in the manuscript ($C=M=10$, $Fe=4000$). A recently submitted manuscript provides more details on the complete route to the chaos of the 2D electroconvection [107]. However, determining the electric Hopf-Rayleigh number of the reduced-order model may require linear stability analysis of the dynamic system, which is beyond the scope of this paper and can be a subject of future work. The following statement is added to the manuscript:

Conclusion and discussion: Similar to the Hopf-Rayleigh number of the Lorenz system [116], the electric Hopf-Rayleigh number determining the onset of electroconvection chaos can be identified by a linear stability analysis of the nonlinear system investigated here.

Reference (numeric same as the main text):

[13] Glukhovsky A, Dolzhansky F. 1980 Three component models of convection in a rotating fluid. *Izv. Acad. Sci. USSR. Atmos. Oceanic Phys* 16, 311–318.

[14] Leonov G. 2015 Existence criterion of homoclinic trajectories in the Glukhovsky–Dolzhansky system. *Physics Letters A* 379, 524–528.

- [15] Chen G, Kuznetsov NV, Leonov GA, Mokaev TN. 2017 Hidden attractors on one path: Glukhovskiy–Dolzhansky, Lorenz, and Rabinovich systems. *International Journal of Bifurcation and Chaos* 27, 1750115.
- [16] Siddheshwar P, Sushma T. 2021 Reduction of a Tri-Modal Lorenz Model of Ferrofluid Convection to a Cubic–Quintic Ginzburg–Landau Equation Using the Center Manifold Theorem. *Differential Equations and Dynamical Systems* pp. 1–19
- [107] Wang Q, Guan Y, Huang J, Wu J. 2021 Chaotic electro-convection flow states of a dielectric liquid between two parallel electrodes. *European Journal of Mechanics- B/Fluids*.
- [116] Kanchana C, Siddheshwar P, Yi Z. 2020 The effect of boundary conditions on the onset of chaos in Rayleigh–Bénard convection using energy-conserving Lorenz models. *Applied Mathematical Modelling* 88, 349–366